# Impact of blood glucose abnormalities on outcomes and disease severity in patients with severe sepsis: An analysis from a multicenter, prospective survey of severe sepsis

Shigeki Kushimoto[1]*, Toshikazu Abe[2,3], Hiroshi Ogura[4], Atsushi Shiraishi[5], Daizoh Saitoh[6], Seitaro Fujishima[7], Toshihiko Mayumi[8], Toru Hifumi[9], Yasukazu Shiino[10], Taka-aki Nakada[11], Takehiko Tarui[12], Yasuhiro Otomo[13], Kohji Okamoto[14], Yutaka Umemura[4], Joji Kotani[15], Yuichiro Sakamoto[16], Junichi Sasaki[17], Shin-ichiro Shiraishi[18], Kiyotsugu Takuma[19], Ryosuke Tsuruta[20], Akiyoshi Hagiwara[21], Kazuma Yamakawa[22], Tomohiko Masuno[23], Naoshi Takeyama[24], Norio Yamashita[25], Hiroto Ikeda[26], Masashi Ueyama[27], Satoshi Fujimi[22], Satoshi Gando[28,29], on behalf of JAAM FORECAST group[¶]

1 Division of Emergency and Critical Care Medicine, Tohoku University Graduate School of Medicine, Sendai, Japan, 2 Department of General Medicine, Juntendo University, Bunkyo, Japan, 3 Health Services Research and Development Center, University of Tsukuba, Tsukuba, Japan, 4 Department of Traumatology and Acute Critical Medicine, Osaka University Graduate School of Medicine, Osaka, Japan, 5 Emergency and Trauma Center, Kameda Medical Center, Kamogawa, Japan, 6 Division of Traumatology, Research Institute, National Defense Medical College, Tokorozawa, Japan, 7 Center for General Medicine Education, Keio University School of Medicine, Shinjuku, Japan, 8 Department of Emergency Medicine, School of Medicine, University of Occupational and Environmental Health, Kitakyushu, Japan, 9 Department of Emergency and Critical Care Medicine, St. Luke's International Hospital, Chuo, Japan, 10 Department of Acute Medicine, Kawasaki Medical School, Kurashiki, Japan, 11 Department of Emergency and Critical Care Medicine, Chiba University Graduate School of Medicine, Chuo-ku, Japan, 12 Department of Trauma and Critical Care Medicine, Kyorin University School of Medicine, Mitaka, Japan, 13 Trauma and Acute Critical Care Center, Medical Hospital, Tokyo Medical and Dental University, Bunkyō, Japan, 14 Department of Surgery, Center for Gastroenterology and Liver Disease, Kitakyushu City Yahata Hospital, Kitakyushu, Japan, 15 Department of Disaster and Emergency Medicine, Kobe University Graduate School of Medicine, Kobe, Japan, 16 Emergency and Critical Care Medicine, Saga University Hospital, Saga, Japan, 17 Department of Emergency and Critical Care Medicine, Keio University School of Medicine, Shinjuku, Japan, 18 Department of Emergency and Critical Care Medicine, Aizu Chuo Hospital, Aizuwakamatsu, Japan, 19 Emergency & Critical Care Center, Kawasaki Municipal Kawasaki Hospital, Kawasaki, Japan, 20 Advanced Medical Emergency & Critical Care Center, Yamaguchi University Hospital, Ube, Japan, 21 Center Hospital of the National Center for Global Health and Medicine, Shinjuku, Japan, 22 Division of Trauma and Surgical Critical Care, Osaka General Medical Center, Osaka, Japan, 23 Department of Emergency and Critical Care Medicine, Nippon Medical School, Bunkyo, Japan, 24 Advanced Critical Care Center, Aichi Medical University Hospital, Nagakute, Japan, 25 Advanced Emergency Medical Service Center Kurume University Hospital, Kurume, Japan, 26 Department of Emergency Medicine, Teikyo University School of Medicine, Itabashi, Japan, 27 Department of Trauma, Critical Care Medicine, and Burn Center, Japan Community Healthcare Organization, Chukyo Hospital, Nagoya, Japan, 28 Division of Acute and Critical Care Medicine, Hokkaido University Graduate School of Medicine, Sapporo, Japan, 29 Department of Acute and Critical Care Medicine, Sapporo Higashi Tokushukai Hospital, Sapporo, Japan

¶ Membership of the JAAM FORECAST group is provided in the Acknowledgments.
* kussie@emergency-medicine.med.tohoku.ac.jp



Data Availability Statement: Data cannot be shared publicly because the collected data contain potentially sensitive information. The data that

support the findings of this study are available from the corresponding author upon reasonable request through the Japanese Association for Acute Medicine for researchers who meet the criteria for access to confidential data. Alternatively, data are available from the Japanese Association for Acute Medicine Ethics Committee with the following contact information: E-mail: jaam-6@bz04.plala.or.jp, name of dataset: JAAM FORECAST-Sepsis.

**Funding:** This study was supported by the Japanese Association for Acute Medicine (2014–01).

**Competing interests:** The authors have declared that no competing interests exist.

**Abbreviations:** ADL, Activities of daily living; APACHE II, Acute Physiologic Assessment and Chronic Health Evaluation II; BMI, Body mass index; CCI, Charlson comorbidity index; CVP, Central venous pressure; ED, Emergency department; EGDT, Early goal direct therapy; FIO2, Fraction of inspired oxygen; FORECAST, Focused Outcomes Research in Emergency Care in Acute Respiratory Distress Syndrome, Sepsis, and Trauma; ICU, Intensive care unit; IQR, Interquartile range; LOS, Length of hospital stay; MAP, Mean arterial pressure; PaO2, Partial pressure of arterial oxygen; ScvO2, Central venous oxygen saturation; SIRS, Systemic inflammatory response syndrome; SOFA, Sepsis-related Organ Failure Assessment; SSC, Surviving Sepsis Campaign; SSCG, Surviving Sepsis Campaign Guidelines; UMIN-CTR, University Hospital Medical Information Network Clinical Trials Registry; VFDs, Ventilator-free days.

# Abstract

## Background

Dysglycemia is frequently observed in patients with sepsis. However, the relationship between dysglycemia and outcome is inconsistent. We evaluate the clinical characteristics, glycemic abnormalities, and the relationship between the initial glucose level and mortality in patients with sepsis.

## Methods

This is a retrospective sub-analysis of a multicenter, prospective cohort study. Adult patients with severe sepsis (Sepsis-2) were divided into groups based on blood glucose categories (<70 (hypoglycemia), 70–139, 140–179, and ≥180 mg/dL), according to the admission values. In-hospital mortality and the relationship between pre-existing diabetes and septic shock were evaluated.

## Results

Of 1158 patients, 69, 543, 233, and 313 patients were categorized as glucose levels <70, 70–139, 140–179, ≥180 mg/dL, respectively. Both the Acute Physiological and Chronic Health Evaluation II and Sequential Organ Failure Assessment (SOFA) scores on the day of enrollment were higher in the hypoglycemic patients than in those with 70–179 mg/dL. The hepatic SOFA scores were also higher in hypoglycemic patients. In-hospital mortality rates were higher in hypoglycemic patients than in those with 70–139 mg/dL (26/68, 38.2% vs 43/221, 19.5%). A significant relationship between mortality and hypoglycemia was demonstrated only in patients without known diabetes. Mortality in patients with both hypoglycemia and septic shock was 2.5-times higher than that in patients without hypoglycemia and septic shock.

## Conclusions

Hypoglycemia may be related to increased severity and high mortality in patients with severe sepsis. These relationships were evident only in patients without known diabetes. Patients with both hypoglycemia and septic shock had an associated increased mortality rate.

# Introduction

Glycemic abnormalities are frequently observed in patients with sepsis. While it is well accepted that hyperglycemia is one of the physiological responses related to acute stress [1–3], a hypoglycemic response may indicate a non-physiological, or pathological, response. Inflammatory mediators and stress hormone-induced glucose utilization is usually associated with increased glucose production. Hypoglycemia develops when the latter fails to respond to the former, secondary to cytokine-induced inhibition of gluconeogenesis in the setting of glycogen depletion [4, 5]. Previous reports have shown that mortality rates are higher in patients with hypoglycemic responses to sepsis [6–8]. However, this relationship is inconsistent and may vary individually based on the presence of diabetes, the severity and etiology of the septic

condition [1, 9–11]. Further, the majority of previous studies have evaluated general intensive care unit (ICU) patients [3, 8, 12, 13] and performed analyses on databases consisting of various infectious diseases, in addition to severe sepsis. Additionally, the majority of these previous works was conducted prior to the publication of the Surviving Sepsis Campaign Guidelines 2012 [14]. This includes revisions of the 6 hour resuscitation bundle and the 24 hour management bundle to the 3-hour/6-hour severe sepsis bundles.

The objective of this study was to evaluate the clinical characteristics, associated glycemic abnormalities, and the relationship between the initial blood glucose level and mortality rates in patients with severe sepsis.

## Materials and methods

### Ethical approval

The study protocol was reviewed and approved by the ethics committee of all participant institutes in the Japanese Association for Acute Medicine (JAAM) study group, Japan. (IRB number 014–0306 on Hokkaido University, the representative for FORECAST). The data collection was performed as a part of routine clinical workup without any interventions, and data management and statistical analyses were processed anonymously. For these reasons, the need for informed consent was waived by the ethics committee/institutional review board.

### Design and setting

The Focused Outcome Research on Emergency Care for Acute Respiratory Distress Syndrome, Sepsis and Trauma (FORECAST) study described the incidence, clinical characteristics, and evolving management of sepsis in Japan [15]. This study is a sub-analysis performed using a cohort of patients with severe sepsis from the FORECAST study. It is comprised of a multicenter, prospective cohort of acutely ill patients; including those with acute respiratory distress syndrome, sepsis, and trauma. The FORECAST study used consecutive patients' data from 59 ICUs in Japan and was conducted from January 2016 to March 2017. Design and reporting of the study adheres to Strengthening the Reporting of Observational Studies in Epidemiology statement [16].

### Participants

The Japanese Association for Acute Medicine Sepsis Registry study group investigated the epidemiology of severe sepsis in patients admitted to 15 ICUs in Japan in 2011, subsequently reporting their findings in 2014 [17]. This registry was implemented with the Surviving Sepsis Campaign Registry, in which Sepsis-2 was used as an inclusion criterion. Since the FORECAST study was planned before the publication of the Sepsis-3 definition, and the purpose of the study includes examining larger comprehensive follow-up reports on the incidence, clinical characteristics, and evolving management of sepsis in Japan, we used the Sepsis-2 definition as the inclusion criterion.

The FORECAST study included adult patients (aged ≥16 years) with severe sepsis based on the Sepsis-2 criteria in 2003 [18]. All patients were admitted to ICUs. Inclusion criteria were: diagnosis of or suspected new onset of infection by the history of present illness; ≥2 systemic inflammatory response syndrome (SIRS) criteria [19]; and presence of at least one organ dysfunction. Sepsis-2 criteria also included: a systolic blood pressure <90 mmHg, mean arterial pressure (MAP) <65 mmHg or a decrease in blood pressure of >40 mmHg, serum creatinine level >2.0 mg/dL or diuresis of <0.5 mL/kg/h, total bilirubin >2.0 mg/dL, platelet count <100,000 cells/mm$^3$, arterial lactate level >2 mmol/L, international normalized ratio >1.5,

and the presence of arterial hypoxemia ($PaO_2$/$FIO_2$ <200 with pneumonia or $PaO_2$/$FIO_2$ <250 without pneumonia) [18]. Exclusion criteria included limitations of sustained life-care or post-cardiopulmonary arrest resuscitation status at the time of the diagnosis of sepsis. This sub-study selected all patients registered in the FORECAST sepsis study and in the sub-analysis. Patients without a value for initial glucose level were excluded.

## Data collection

Data were obtained from the FORECAST database, which was compiled by the FORECAST investigators. Patient data included such information as patient demographics, admission source, comorbidities, suspected site(s) of infection, organ dysfunction(s), and sepsis-related severity scores. The Sequential Organ Failure Assessment (SOFA) score was calculated using physiological and laboratory values during the initial evaluation. We also obtained data on compliance with established sepsis care protocols, such as serum lactate levels obtained within three hours. Data collection was performed as a part of the routine clinical workup.

Blood glucose levels were measured using a blood gas analyzer (and not a glucometer) as recommend by Surviving Sepsis Campaign: International Guidelines for Management of Severe Sepsis and Septic Shock: 2012 [14]. All glucose levels were measured prior to the administration of corticosteroids (if required).

In-hospital mortality, 28-day mortality, disposition after discharge, and number of ICU-free and ventilator-free days (VFDs) were used as outcome measures.

## Data definitions

Based on the initial blood glucose value, patients were divided into four groups: <70 (defined as hypoglycemia), 70–139, 140–179, ≥180 mg/dL. Although there is no universally agreed upon definition of hypoglycemia, a blood glucose level of 70 mg/dL or less is widely accepted as the definition of hypoglycemia [20]. As such, it is the classification defined as "hypoglycemia" in this study. We defined all glucose categories a priori based on previously published studies, regardless of the presence or absence of diabetes [21]. Although inpatient management was performed based on the Surviving Sepsis Campaign Guidelines 2012, no uniform protocol to control blood glucose level was used in this study.

Septic shock was defined by the Sepsis-2 criteria [18]. In the evaluation of organ dysfunction, hypotension was defined as a systolic blood pressure <90 mmHg, MAP<65 mmHg, or a decrease in blood pressure >40 mmHg. Acute lung injury included arterial hypoxemia; that is, $PaO_2$/$FIO_2$ <200 mmHg with pneumonia or $PaO_2$/$FIO_2$ <250 mmHg without pneumonia. The results of the Charlson comorbidity index were classified into four previously defined grades of severity: 0 (none), 1–2 (low), 3–4 (moderate), and ≥5 points (high) [22]. We also measured the compliance with the bundles proposed in the Surviving Sepsis Campaign Guidelines (SSCG) 2012 [14]. We defined compliance as evidence that all bundle elements were adhered to within the established time-frame (i.e., 3 h or 6 h) and to the respective indications (i.e., septic shock or lactate >4 mmol/L). In addition, VFDs were defined as the number of days on which a patient was able to breathe without a ventilator during the initial 28 days after enrollment. The number of VFDs of patients who died during the study period was assigned as 0. The number of ICU-free days was calculated in this same manner.

## Hepatic disease/dysfunction, glycemic abnormalities, Sepsis-3, and outcomes

To evaluate the relationship between hepatic impairment and glycemic abnormalities, the presence of moderate to severe hepatic disease as a comorbidity and hepatic SOFA score were

compared among groups. In addition, to assess the influence of hepatic impairment on the outcome of hypoglycemia, the presence of moderate to severe hepatic disease and hepatic SOFA score >0 were compared between hypoglycemia and non-hypoglycemia patients. Severe, moderate, and mild liver disease as comorbidities were defined as follows: severe; cirrhosis and portal hypertension with variceal bleeding history; moderate, cirrhosis and portal hypertension but no variceal bleeding history; and mild, chronic hepatitis (or cirrhosis without portal hypertension) [23].

To identify patients who were diagnosed with sepsis according to the Sepsis-3 criteria [24], we calculated an acute increase in SOFA scores of ≥2 as follows: a baseline SOFA score of 0 was assumed in patients without a diagnosis of any chronic disease, as defined per APACHE II scores. If a chronic disease was present, a baseline SOFA score of 2 was assigned (as previously reported [25]).

### Analysis

Descriptive statistics included proportions for categorical and median (interquartile range) for continuous variables, as not all variables had a normal distribution. Since the amount of missing data was low (with the exception of bundle data), no assumptions were made for missing data.

Categorical variables were summarized using proportions and compared using Fisher's exact test or chi-square tests. Kruskal-Wallis one-way analysis of variance was used to compare results among multiple groups. Odds ratios are reported relative to a reference range of blood glucose. Kaplan–Meier curves for patient survival was used to assess the duration of survival, and compared by using a log-rank test.

We assessed the relationships between hospital mortality and the various independent variables by a Cox regression model. Mortality was used as the criterion variable (death = 1; survival = 0), while age ≤75 years or not, Charlson comorbidity index, SOFA and Acute Physiological and Chronic Health Evaluation II (APACHE II) scores, and blood glucose level <70 mg/dL or not were used as explanatory variables. Before the multivariate analysis, the variance inflation factor for each explanatory variable was calculated. For all explanatory variables, the variance inflation factor was less than five. After an initial review of the data demonstrated a significant association between blood glucose (<70 mg/dL or not) and survival duration, we performed a sensitivity analysis that changed the Charlson comorbidity index, SOFA score, and age >75 years with the APACHE II score. We defined statistical significance as P<0.05 for single comparisons and P<0.00833 for multiple comparisons (after Bonferroni correction). Statistical analyses were performed using SPSS software, version 25.0 (IBM, Armonk, NY, USA).

## Results

### Baseline characteristics

A total of 1,184 patients with severe sepsis were included in the sepsis cohort of the FORE-CAST study. Of those, 26 patients with missing data on blood glucose at the time of admission were excluded from this analysis. The patients were divided into four groups based on the initial blood glucose value (Fig 1).

Patient characteristics at enrollment are shown in Tables 1 and S1. The median age was 73 years, and the majority of the infections were of pulmonary (31.0%), intra-abdominal (26.2%), and urinary (18.7%) origin. Positive blood culture was observed in 58.8% of patients, and 62.8% of patients were diagnosed with septic shock.

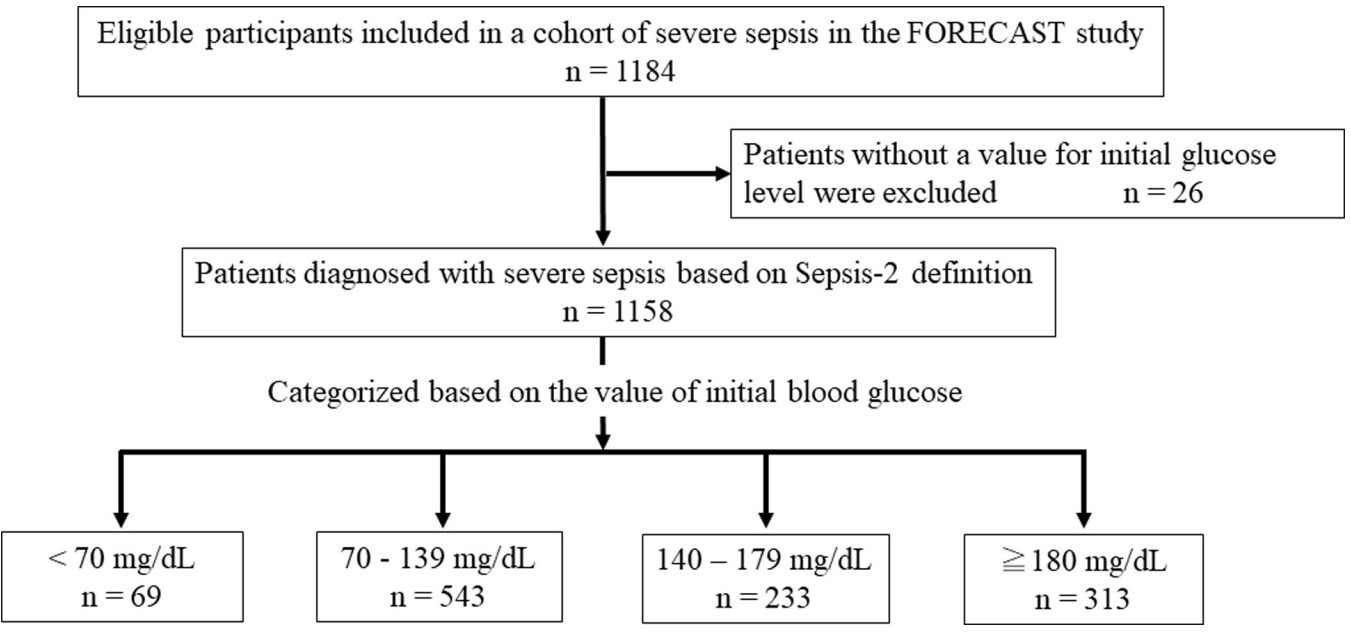

**Fig 1. Flow chart of subject selection and blood glucose categories.**

Of 1,158 enrolled patients, 69, 543, 233, and 313 patients were categorized as having glucose levels <70, 70–139, 140–179, ≥180 mg/dL, respectively. Hypoglycemia was observed in 6.0% of all patients. The incidence of moderate to severe liver disease and the prevalence of septic shock were significantly higher in patients with blood glucose level <70 mg/dL than in all other groups.

## Blood glucose and outcomes

Although the number of ICU-free days and VFDs were not significantly different between the groups, the in-hospital and 28-day mortality rates were significantly higher in patients with blood glucose level <70 mg/dL than in patients of all other groups (Table 2).

Outcome data for patients with sepsis according to the Sepsis-3 criteria (S2 Table) and patients divided according to quartiles of glucose level (S3 Table) and the odds ratios for in-hospital mortality (relative to patients with blood glucose levels of 70–139 mg/dL) (S4 Table) are shown as supplementary tables.

Kaplan–Meier curves analyzing patient survival demonstrated a significant difference between patients with glucose level <70 mg/dL and all other groups (S1 Fig).

## Influence of hepatic disease/dysfunction on mortality

The hepatic SOFA scores in the patients who were categorized according to their glucose levels of <70, 70–139, 140–179, and >180 mg/dL were 1 (0–1), 0 (0–1), 0 (0–1), and 0 (0–1), respectively (p<0.001). The score was significantly higher in the blood glucose level <70 mg/dL group than in the other groups (Table 1).

In contrast, the presence of "moderate to severe liver disease" as a comorbidity was not different between hypoglycemia and non-hypoglycemia patients (8/69, 151/1089, p = 0.719). Furthermore, hepatic SOFA score on admission was not higher in survivors than in non-survivors (p = 0.232). Odds ratios of hepatic SOFA score of ≥1 for in-hospital mortality and 28-day mortality were 1.18 (0.89–1.57) and 1.22 (0.90–1.66), respectively.

**Table 1. Characteristics and blood glucose levels (mg/dL) on admission among patients with severe sepsis in 59 intensive care units in Japan (n = 1158).**

| Characteristics | All patients (n = 1158) | <70 mg/dL (n = 69) | 70–139 mg/dL (n = 543) | 140–179 mg/dL (n = 233) | ≥180 mg/dL (n = 313) | P value |
|---|---|---|---|---|---|---|
| Age | 73 (64–81) | 73 (66–79.5) | 72 (62–82) | 74 (64–81.5) | 74 (64.5–81) | 0.292 |
| Male sex (n, %) | 696, 60.1% | 34, 49.3%**,# | 313, 57.6% | 149, 63.9% | 200, 63.9% | 0.045 |
| BMI | 21.8 (19.0–24.7) | 21.3 (19.1–24.8) | 21.3 (18.8–24.2) | 21.9 (19.5–25.1) | 22.4 (19.1–25.4) | 0.076 |
| Pre-existing diabetes mellitus (n, %) | 270, 23.3% | 13, 18.6%# | 69, 12.7%# | 45, 19.3%# | 143, 45.7% | <0.001 |
| Coexisting conditions (n, %) | | | | | | |
| Myocardial infarction | 57, 4.9% | 1, 1.4% | 26, 4.8% | 10, 4.3% | 20, 6.4% | 0.329 |
| Congestive heart failure | 124, 10.7% | 5, 7.2% | 52, 9.6% | 36, 15.5% | 31, 9.9% | 0.063 |
| Cerebrovascular disease | 136, 11.7% | 5, 4.2% | 42, 13.3% | 26, 11.2% | 33, 10.5% | 0.382 |
| COPD | 81, 7.0% | 4, 5.8% | 38, 7.0% | 19, 8.2% | 20, 6.4% | 0.847 |
| Connective tissue disease | 83, 7.2% | 6, 8.7% | 48, 8.8% | 13, 5.6% | 16, 5.1% | 0.145 |
| Peptic ulcer disease | 32, 2.8% | 1, 1.4% | 15, 2.8% | 5, 2.1% | 11, 3.5% | 0.698 |
| Diabetes mellitus without organ damage | 196, 16.9% | 6, 8.7%**,# | 46, 8.5%**,# | 37, 15.9% | 101, 34.2% | <0.001 |
| Diabetes mellitus with organ damage | 74, 6.4% | 7, 10.1%*,** | 23, 4.2%# | 8, 3.4%# | 36, 11.5% | <0.001 |
| Chronic kidney disease | 83, 7.2% | 4, 5.8% | 40, 7.4% | 20, 8.6% | 19, 6.1% | 0.683 |
| Malignancy (solid) | 159, 13.7% | 8, 11.5% | 69, 12.7% | 42, 18.0% | 40, 12.8% | 0.202 |
| Moderate to severe liver disease | 26, 2.2% | 8, 11.6%*,**,# | 9, 1.7% | 2, 0.9% | 7, 2.2% | <0.001 |
| CCI | 1(0–2) | 1(0–2) | 1(0–2) | 1(0–2) | 1(0–3) | 0.293 |
| ADL: Inactive (n, %) | 282, 24.4% | 10, 14.5% | 139, 25.6% | 60, 25.8% | 73, 23.3% | 0.003 |
| Suspected site of infection (n, %) | | | | | | 0.001 |
| Lung | 359, 31.0% | 20, 29.0% | 140, 25.8% | 82, 35.2% | 117, 37.4% | |
| Abdomen | 303, 26.2% | 17, 24.6% | 159, 29.3% | 62, 26.6% | 65, 20.8% | |
| Urinary tract | 216, 18.7% | 15, 23.2% | 107, 19.7% | 43, 18.5% | 50, 16.0% | |
| Soft tissue | 115, 9.9% | 5, 7.2% | 66, 12.2% | 14, 6.0% | 30, 9.6% | |
| Positivity of blood cultures (n, %) | 681, 58.8% | 43, 62.3% | 338, 62.2% | 117, 50.2% | 183, 58.5% | 0.072 |
| Septic shock (n, %) | 727, 62.8% | 57, 82.6%*,**,# | 351, 64.6% | 132, 56.7% | 187, 59.7% | 0.001 |
| Lactate level (mmol/L) | 3.00 (1.80–5.30) | 5.50 (3.05–9.75) | 2.70 (1.60–5.10) | 2.80 (1.78–4.93) | 3.25 (2.18–5.60) | <0.001 |
| qSOFA score | 2 (1–2) | 2 (1–2) | 2 (1–2) | 2 (1–2) | 1 (1–2) | 0.003 |
| APACHE II score | 23 (17–29) | 29(21.75–36) *,** | 22 (16–29) | 22 (15–28) | 23(17–30) | 0.013 |
| SIRS score | 3 (2–4) | 3 (2–3.75) | 3 (2–3.5) | 3 (2–4) | 3 (3–4) | 0.032 |
| SOFA score | 9 (6–11.25) | 11.5 (9–14) *,** | 8(6–11) | 8(5–11) | 9 (6–11) | <0.001 |
| Compliance with all applicable elements of sepsis 3-h bundle (n, %) | | | | | | |
| Entire 3-h resuscitation bundle† | 535, 64.5% | 47, 75.8% | 240, 62.5% | 109, 69.9% | 139, 61.2% | 0.067 |
| B1. Serum lactate obtained | 1125, 97.2% | 68, 98.6% | 529, 97.4% | 222, 95.3% | 306, 97.8% | 0.261 |
| B2. Broad-spectrum antibiotic given | 970, 83.8% | 59, 85.5% | 452, 83.2% | 192, 82.4% | 267, 85.6% | 0.721 |
| B3. Blood cultures obtained before broad-spectrum antibiotic administration | 1064, 92.0% | 66, 95.7% | 498, 91.9% | 213, 91.4% | 287, 92.0% | 0.712 |
| B4. 30 mg/kg crystalloid fluid bolus delivered (yes/ cases with indication)† | 636, 76.6% | 51, 82.3% | 295, 76.6% | 126, 80.8% | 164, 72.3% | 0.170 |
| Number of patients with corticosteroid requirement within 6 h (n, %) | 191, 16.5% | 30, 43.5%*,**,# | 90, 16.6% | 24, 10.3% | 47, 15.0% | <0.001 |

Reported counts (proportions) for categorical and median (interquartile range) for continuous variables.

†septic shock or lactate >4 mmol/L.

Missing data: BMI = 20, Admission source = 2, ADL = 1, blood culture = 6, qSOFA = 24, APACHE II = 124, SIRS = 0, SOFA = 153

ICU = intensive care unit, BMI = body mass index, ED = emergency department, CCI = Charlson Comorbidity Index, COPD = chronic obstructive pulmonary disease; AIDS = acquired immune deficiency syndrome, ADL = activities of daily living, IV = intravenous, qSOFA = quick sepsis organ failure assessment, APACHE = acute physiology and chronic health evaluation, SIRS = systemic inflammatory response syndrome, SOFA = sequential organ failure assessment, INR = international normalized ratio.

*, <0.00833 vs. 70–139 mg/dL.

**, <0.00833 vs. 140–179 mg/dL.

#, <0.00833 vs. ≥180 mg/dL.

**Table 2. Clinical outcomes and blood glucose on admission.**

| Outcomes | All patients (n = 1158) | <70 mg/dL (n = 69) | 70–139 mg/dL (n = 543) | 140–179 mg/dL (n = 233) | ≥180 mg/dL (n = 313) | |
|---|---|---|---|---|---|---|
| In-hospital mortality | 266/1127, 23.6% | 26/68, 38.2%*,** | 123/533, 23.1% | 43/221, 19.5% | 54/301, 24.6% | 0.002 |
| 28-day mortality | 213/1116, 19.1% | 24/68, 35.3%*,** | 99/529, 18.7% | 36/218, 16.5% | 54/301, 24.6% | 0.002 |
| Survivor dispositions | (n = 861) | (n = 42) | (n = 410) | (n = 178) | (n = 231) | 0.025 |
| Home (n, %) | 317, 36.8% | 8, 19.0% | 158, 38.5% | 72, 40.4% | 79, 34.2% | |
| Transfer (n, %) | 544, 63.2% | 34, 81.0% | 252, 61.5% | 106, 59.6% | 152, 65.8% | |
| ICU-free days | 19 (11–24) | 16.5 (8.5–22) | 22 (12–24) | 20 (10–24) | 18 (9–23) | 0.048 |
| Ventilator-free days | 21 (0–28) | 14 (0–25) | 22 (0–28) | 21 (0.75–28) | 19 (0–26) | 0.051 |
| Length of hospital stay | 24 (12–46) | 23.5 (7.25–43.5) | 23 (11–45.5) | 25 (14–39) | 24 (13–51) | 0.925 |

Reported counts (proportions) for categorical and median (interquartile range) for continuous variables.

Missing data: in-hospital mortality = 31, 28-day mortality = 42, ICU-free days = 255, Ventilator-free days = 42, Length of hospital stay = 31.

ICU = intensive care unit.

*, <0.00833 vs. 140–179 mg/dL.

**, <0.00833 vs. ≥180 mg/dL.

## Blood glucose level and clinical outcomes in patients with known diabetes and septic shock

The in-hospital mortality rates with each group with or without a diabetes diagnosis were compared. Although a significantly higher in-hospital mortality rate was observed in patients with blood glucose levels <70 mg/dL than in those with blood glucose levels of 140–179 mg/dL among patients without known diabetes, this difference was not evident in patients with known diabetes (Table 3).

Odds ratio for the in-hospital mortality ratio with each group with or without a diabetes diagnosis were also analyzed, relative to the group with a normal glycemic reference range (70–139 mg/dL) (S4 Table, middle portion). The relationship between mortality and blood glucose level was only significant in patients without known diabetes and blood glucose level <70 mg/dL (odds ratio, 2.07, 95% confidence interval, 1.15–3.72). Significant relationships were not demonstrated in patients with or without septic shock (Tables 3 and S4, lower part).

Mortality in patients with both hypoglycemia and septic shock was 2.5-times higher than that in patients without hypoglycemia and septic shock (15.8% vs. 39.3%) (Table 4), and the unadjusted odds ratio for the in-hospital mortality rate in patients with presence of both blood glucose level <70 mg/dL and septic shock was 3.44 (95% confidence interval, 1.90–6.24) when compared to non-hypoglycemic patients (blood glucose ≥70 mg/dL) without septic shock (S5 Table).

Evaluation utilizing the Cox regression model demonstrated that in-hospital mortality rates and blood glucose levels <70 mg/dL on admission were independently associated with a survival duration. Sensitivity analysis also demonstrated this same association (Table 5).

## Discussion

Our findings clearly indicate that severe sepsis patients with hypoglycemia, defined as blood glucose levels <70 mg/dL, have substantial divergent characteristics and clinical outcomes when compared to those with normal blood glucose levels. Although some patients with hypoglycemia had an associated increase in mortality, this relationship was only evident in patients without known diabetes mellitus. In addition, patients with both hypoglycemia and septic shock had an associated increase in mortality rate. This study was conducted using a dataset

**Table 3. In-hospital mortality and body glucose category on admission.**

| Glucose categories | Mortality (n, %) |
|---|---|
| Patients with known diabetes mellitus | |
| <70 mg/dL | 6/13, 46.2% |
| 70–139 mg/dL | 22/66, 33.3% |
| 140–179 mg/dL | 12/42, 28.6% |
| ≥180 mg/dL | 32/140, 22.9% |
| Patients without known diabetes mellitus | |
| <70 mg/dL | 20/55, 36.4%* |
| 70–139 mg/dL | 101/467, 21.6% |
| 140–179 mg/dL | 31/179, 17.3% |
| ≥180 mg/dL | 42/165, 25.5% |
| Patients with septic shock | |
| <70 mg/dL | 22/56, 39.3% |
| 70–139 mg/dL | 94/343, 27.4% |
| 140–179 mg/dL | 26/123, 21.1% |
| ≥180 mg/dL | 55/182, 30.2% |
| Patients without septic shock | |
| <70 mg/dL | 4/12, 33.3% |
| 70–139 mg/dL | 29/190, 15.3% |
| 140–179 mg/dL | 17/98, 17.3% |
| ≥180 mg/dL | 19/123, 15.4% |

*, <0.00833 vs. 140–179 mg/dL

**Table 4. In-hospital mortality in patients with or without hypoglycemia (<70 mg/dL) and septic shock.**

| | Mortality (n, %) |
|---|---|
| Glucose <70 mg/dL with Septic shock | 22/56, 39.3%* |
| Glucose <70 mg/dL without Septic shock | 4/12, 33.3% |
| Glucose ≥70 mg/dL with Septic shock | 175/648, 27.0% |
| Glucose ≥70 mg/dL without Septic shock | 65/411, 15.8% |

*, <0.00833 vs. Glucose >70 mg/dL without Septic shock

**Table 5. Cox regression analysis for the duration of survival (forced entry model).**

| | Hazard ratio | 95.0% CI | | P value |
|---|---|---|---|---|
| Glucose level <70 mg/dL | 1.38 | 1.03 | 2.18 | 0.044 |
| Charlson Comorbidity Index | 1.15 | 1.08 | 1.23 | < 0.001 |
| SOFA score | 1.13 | 1.00 | 1.70 | 0.051 |
| Age >75 years | 1.30 | 0.58 | 0.94 | < 0.001 |
| Sensitivity analysis | | | | |
| Glucose level <jl70 mg/dL | 1.38 | 1.02 | 2.22 | 0.048 |
| APACHE II score | 1.08 | 1.06 | 1.09 | < 0.001 |

Independent variable = Duration of Survival

Explanatory variables = hypoglycemia (defined as <70 mg/dL), Age >75 years, Charlson Comorbidity Index, APACHE II score, SOFA score

consisting of patients with severe sepsis, specifically to address the recent advances in the management of the most serious septic cases.

## Dysglycemia in patients with sepsis

Several studies have suggested that patients with stress-induced hyperglycemia and no previous diagnosis of diabetes face more dire consequences at a given severity of hyperglycemia than in those with pre-existing diabetes. A retrospective review that stratified patients as having normoglycemia, pre-existing diabetes, or newly diagnosed hyperglycemia demonstrated that mortality was 18.3 times higher in patients with newly diagnosed hyperglycemia [26]. Furthermore, there is a high mortality rate in hyperglycemic patients without known diabetes and an absence of a relationship between hyperglycemia and mortality rates in patients with diabetes in different conditions [9, 10]. Recently, it has been reported that septic ICU patients with insulin-treated diabetes have lower adjusted hospital mortality rates and higher peak blood glucose levels when compared to non-insulin-treated patients. This suggests that septic patients with a pre-existing diagnosis of diabetes present with an altered relationship between hospital mortality and elevated glucose levels [27]. Although the findings are inconsistent, we suggest hyperglycemia may have different clinical implications in patients with or without pre-existing diabetes.

Hypoglycemia has been suggested to be associated with an increased mortality rate in critically ill patients [6–8]. There is a J- or U-shaped relationship between glucose levels and mortality, such that hyper- and hypoglycemic patients have higher mortality rates than normoglycemic patients [28, 29]. The NICE-SUGAR study reported a relationship between hypoglycemia and the risk of death [6]. The mortality rates in patients who did not have hypoglycemia, moderate hypoglycemia, and severe hypoglycemia were 23.5%, 28.5%, and 35.4%, respectively. The adjusted hazard ratio for mortality among patients with severe hypoglycemia was 2.10. In other studies, hypoglycemia has been independently associated with an increased mortality rate [7, 8].

Although different associations between the effects of hyperglycemia and elevated mortality risk in pre-existing diabetes have been suggested [21, 30], we did not observe this association in this study. This may have been a result of the glucose categories used in this study or a lack of a standardized inpatient glucose control protocol. However, patients with hypoglycemia had an associated increase in mortality rate in this study, which is consistent with findings of the majority of previous studies. Interestingly, this relationship was only evident in patients without known diabetes mellitus, which is a novel finding implicating an association between dysglycemia and sepsis only in patients without a preexisting diabetes diagnosis.

The liver plays an important role in glucose metabolism, and hypoglycemia may occur due to impaired ability of the liver to increase plasma glucose through gluconeogenesis [31]. Hypoglycemia has also been reported as a common manifestation of sepsis patients in cirrhosis [32]. Although the hepatic SOFA score in hypoglycemic patients was significantly higher than those in other groups, the presence of "moderate to severe liver disease" as a comorbidity was not different between hypoglycemia and non-hypoglycemia patients in this study. In addition, hepatic SOFA score on admission in survivors was not higher than that in non-survivors, and the odds ratios of hepatic SOFA scores of ≥1 for in-hospital mortality and 28-day mortality were 1.18 (0.89–1.57) and 1.22 (0.90–1.66), respectively. Hypoglycemia may be related to hepatic impairment accompanied by corticosteroid insufficiency, and further study including a larger number of patients is required to elucidate the impact of the relationship between hypoglycemia and hepatic impairment on the outcome of patients with sepsis.

### Hypoglycemia in patients with sepsis—biologic plausibility

Sepsis has been observed to be commonly associated with hypoglycemia [33]. Although spontaneous hypoglycemia is strongly associated with mortality [34], a causal relationship may also be plausible as hypoglycemia can have substantial and varied biological effects in critically ill patients. These effects include an increased systemic inflammatory response, induced neuroglycopenia, inhibition of the corticosteroid response to stress, impairments in the responsiveness and exhaustion of the sympathetic nervous system, and induction of hypotension, vasodilatation, and nitric oxide release [6]. These critical pathological responses suggest hypoglycemia is an epiphenomenon of severe organ dysfunction that can precede death. Although the mechanisms and relationships between hypoglycemia and disease severity in septic patients has not been clarified, inflammatory cytokines, which both increase glucose utilization and inhibit gluconeogenesis [4], may be implicated. That is, hypoglycemia may be a part of a phenotype reflecting a pathological acute stress response.

Hypoglycemia and hypotension are well demonstrated symptoms of adrenal insufficiency. These symptoms in patients with sepsis can be related to Critical Illness-Related Corticosteroid Insufficiency (CIRCI). As such, patients with hypoglycemia may be treated with systemic corticosteroid administration. In this study, the rate of septic shock in hypoglycemic patients was significantly higher, and administration of corticosteroids was more frequent in the hypoglycemic patients. Moreover, the unadjusted odds ratio for in-hospital mortality in patients with both blood glucose <70 mg/dL and presence of septic shock was 3.444 (95% confidence interval, 1.904–6.236) compared to that in patients with blood glucose ≥70 mg/dL without septic shock. Although the prevalence of CIRCI was not evaluated in this study, these relationships may contribute to a poor outcome in patients with hypoglycemia, and the efficacy of corticosteroid administration in patients with hypoglycemia, with or without CIRCI, will need further clarification.

This study involved analyses of databases restricted to patients with severe sepsis only, and after the publication of the Surviving Sepsis Campaign Guidelines 2012 [14]. This is different from most other previous studies, which included patients with less severe conditions in their analyses [3, 8, 12, 13]. There are some limitations of our study. First, we may have underestimated the number of patients with diabetes because of our patient inclusion procedure. Previous studies have shown that diabetes is often unrecognized in hospitalized patients [29], and we defined pre-existing diabetes based on the hospital records for Charlson comorbidity indexing. Both the presence of preexisting diabetes and preadmission glucose control status have important implications for patient health and outcome. However, glucose control status, i.e. HgbA1c levels, was not included in this prospective study. This may be a significant limitation in demonstrating a relationship between glycemic abnormalities and the relevant clinical outcomes of patients with sepsis. Second, although we evaluated the relationships between the initial blood glucose levels and mortality rates in patients with severe sepsis, we acknowledge that glucose management may influence patient outcomes substantially. The effect of glucose management was not included in the current analysis, and as such was not considered as a prognostic factor. Third, although we used the blood glucose levels measured during initial evaluation and management, the time between measurement and the diagnosis of severe sepsis or septic shock was not defined in this registry. These measurement related issues may have influenced the results.

## Conclusions

Hypoglycemia may be related to increased disease severity and a higher mortality rate in patients with sepsis and an independently associated predictor of poor clinical outcome. These

relationships were evident only in patients without known diabetes. Patients with both hypoglycemia and septic shock had an associated increase in mortality rates. Although hypoglycemia may be related to hepatic impairment, further studies are needed to examine this relationship with regard to the outcomes of patients with sepsis.

## Supporting information

**S1 Table. Characteristics and blood glucose levels (mg/dL) on admission among patients with severe sepsis in 59 intensive care units in Japan (n = 1158).**
(DOCX)

**S2 Table. Clinical outcomes and blood glucose levels at admission in patients diagnosed with sepsis according to the Sepsis-3 criteria.**
(DOCX)

**S3 Table. Clinical outcomes according to the quartiles of patient's blood glucose levels at admission.**
(DOCX)

**S4 Table. In-hospital mortality and body glucose category on admission.**
(DOCX)

**S5 Table. In-hospital mortality in patients with or without hypoglycemia (<70 mg/dL) and septic shock.**
(DOCX)

**S6 Table. List of the names of all of the ethics committees that reviewed and approved this study.**
(DOCX)

**S1 Fig.**
(TIF)

## Acknowledgments

We thank the JAAM FORECAST Study Group for contribution to this study.

Investigators of the JAAM FORECAST Study Group:

Nagasaki University Hospital (Osamu Tasaki); Osaka City University Hospital (Yasumitsu Mizobata); Tokyobay Urayasu Ichikawa Medical Center (Hiraku Funakoshi); Aso Iizuka Hospital (Toshiro Okuyama); Tomei Atsugi Hospital (Iwao Yamashita); Hiratsuka City Hospital (Toshio Kanai); National Hospital Organization Sendai Medical Center (Yasuo Yamada); Ehime University Hospital (Mayuki Aibiki); Okayama University Hospital (Keiji Sato); Tokuyama Central Hospital (Susumu Yamashita); Fukuyama City Hospital (Susumu Yamashita); JA Hiroshima General Hospital (Kenichi Yoshida); Kumamoto University Hospital (Shunji Kasaoka); Hachinohe City Hospital (Akihide Kon); Osaka City General Hospital (Hiroshi Rinka); National Hospital Organization Disaster Medical Center (Hiroshi Kato); University of Toyama (Hiroshi Okudera); Sapporo Medical University (Eichi Narimatsu); Okayama Saiseikai General Hospital (Toshifumi Fujiwara); Juntendo University Nerima Hospital (Manabu Sugita); National Hospital Organization Hokkaido Medical Center (Yasuo Shichinohe); Akita University Hospital (Hajime Nakae); Japanese Red Cross Society Kyoto Daini Hospital (Ryouji Iiduka); Maebashi Red Cross Hospital (Mitsunobu Nakamura); Sendai City Hospital (Yuji Murata); Subaru Health Insurance Society Ota Memorial Hospital (Yoshitake Sato); Fukuoka University Hospital (Hiroyasu Ishikura); Ishikawa Prefectural Central Hospital

(Yasuhiro Myojo); Shiga University of Medical Science (Yasuyuki Tsujita); Nihon University School of Medicine (Kosaku Kinoshita); Seirei Yokohama General Hospital (Hiroyuki Yama-guchi); National Hospital Organization Kumamoto Medical Center (Toshihiro Sakurai); Sai-seikai Utsunomiya Hospital (Satoru Miyatake); National Hospital Organization Higashi-Ohmi General Medical Center (Takao Saotome); National Hospital Organization Mito Medical Center (Susumu Yasuda); Tsukuba Medical Center Hospital (Toshikazu Abe); Osaka University Graduate School of Medicine (Hiroshi Ogura, Yutaka Umemura); Kameda Medical Center (Atsushi Shiraishi); Tohoku University Graduate School of Medicine (Shigeki Kushimoto); National Defense Medical College (Daizoh Saitoh); Keio University School of Medicine (Sei-taro Fujishima, Junichi Sasaki); University of Occupational and Environmental Health (Toshi-hiko Mayumi); Kawasaki Medical School (Yasukazu Shiino); Chiba University Graduate School of Medicine (Taka-aki Nakada); Kyorin University School of Medicine (Takehiko Tarui); Kagawa University Hospital (Toru Hifumi); Tokyo Medical and Dental University (Yasuhiro Otomo); Hyogo College of Medicine (Joji Kotani); Saga University Hospital (Yui-chiro Sakamoto); Aizu Chuo Hospital (Shin-ichiro Shiraishi); Kawasaki Municipal Kawasaki Hospital (Kiyotsugu Takuma); Yamaguchi University Hospital (Ryosuke Tsuruta); Center Hospital of the National Center for Global Health and Medicine (Akiyoshi Hagiwara); Osaka General Medical Center (Kazuma Yamakawa); Aichi Medical University Hospital (Naoshi Takeyama); Kurume University Hospital (Norio Yamashita); Teikyo University School of Medicine (Hiroto Ikeda); Rinku General Medical Center (Yasuaki Mizushima); Hokkaido University Graduate School of Medicine (Satoshi Gando).

## Author Contributions

**Conceptualization:** Shigeki Kushimoto, Toshikazu Abe, Hiroshi Ogura, Atsushi Shiraishi, Daizoh Saitoh, Seitaro Fujishima, Toshihiko Mayumi, Satoshi Gando.

**Data curation:** Shigeki Kushimoto, Hiroshi Ogura, Atsushi Shiraishi, Daizoh Saitoh, Seitaro Fujishima, Toshihiko Mayumi, Toru Hifumi, Yasukazu Shiino, Taka-aki Nakada, Takehiko Tarui, Yasuhiro Otomo, Kohji Okamoto, Yutaka Umemura, Joji Kotani, Yuichiro Saka-moto, Junichi Sasaki, Shin-ichiro Shiraishi, Kiyotsugu Takuma, Ryosuke Tsuruta, Akiyoshi Hagiwara, Kazuma Yamakawa, Tomohiko Masuno, Naoshi Takeyama, Norio Yamashita, Hiroto Ikeda, Masashi Ueyama, Satoshi Fujimi, Satoshi Gando.

**Formal analysis:** Shigeki Kushimoto, Toshikazu Abe.

**Writing – original draft:** Shigeki Kushimoto, Toshikazu Abe, Seitaro Fujishima.

**Writing – review & editing:** Shigeki Kushimoto, Toshikazu Abe, Hiroshi Ogura, Atsushi Shir-aishi, Daizoh Saitoh, Toshihiko Mayumi, Toru Hifumi, Yasukazu Shiino, Taka-aki Nakada, Takehiko Tarui, Yasuhiro Otomo, Kohji Okamoto, Yutaka Umemura, Joji Kotani, Yuichiro Sakamoto, Junichi Sasaki, Shin-ichiro Shiraishi, Kiyotsugu Takuma, Ryosuke Tsuruta, Akiyoshi Hagiwara, Kazuma Yamakawa, Tomohiko Masuno, Naoshi Takeyama, Norio Yamashita, Hiroto Ikeda, Masashi Ueyama, Satoshi Fujimi, Satoshi Gando.

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
