## [Editor Report · Decision Letter 0]

13 Sep 2019

PONE-D-19-25111

Impact of Blood Glucose Abnormalities on Outcomes and Disease Severity in Patients with Severe Sepsis: An Analysis from a Multicenter, Prospective Survey of Severe Sepsis

PLOS ONE

Dear Prof. Kushimoto,

Thank you for submitting your manuscript to PLOS ONE. After careful consideration, we feel that it has merit but does not fully meet PLOS ONE’s publication criteria as it currently stands. Therefore, we invite you to submit a revised version of the manuscript that addresses the points raised during the review process.

Before your manuscript can be sent out for review, please review the journal's data availability policy (https://journals.plos.org/plosone/s/data-availability) paying particular attention to the following paragraph:

Data made available to all interested researchers upon request

The Data Availability Statement must specify “Data available on request” and identify the group to which requests should be submitted (e.g., a named data access committee or named ethics committee). The reasons for restrictions on public data deposition must also be specified. *Note that it is not acceptable for an author to be the sole named individual responsible for ensuring data access.*

Thank you for your attention to this matter. I look forward to reviewing your manuscript once the necessary changes have been made.

We would appreciate receiving your revised manuscript by Oct 28 2019 11:59PM. To enhance the reproducibility of your results, we recommend that if applicable you deposit your laboratory protocols in protocols.io, where a protocol can be assigned its own identifier (DOI) such that it can be cited independently in the future. For instructions see: http://journals.plos.org/plosone/s/submission-guidelines#loc-laboratory-protocols

We look forward to receiving your revised manuscript.

Kind regards,

Robert Ehrman, MD, MS

Academic Editor

PLOS ONE

Journal Requirements:

1. Please provide additional details regarding participant consent. In the ethics statement in the Methods and online submission information, please ensure that you have specified (1) whether consent was informed and (2) what type you obtained (for instance, written or verbal). If your study included minors, state whether you obtained consent from parents or guardians. If the need for consent was waived by the ethics committee, please include this information.

We additionally ask that you please include a list of the names of all of the ethics committees that reviewed and approved this study. Please include these as a Supporting Information file.
---

## [Author Response · Author response to Decision Letter 0]

17 Sep 2019

Response to the Editor’s comments

We thank the Editor for the favorable comments and acknowledgment of our work. We have revised our manuscript accordingly.

a) The Data Availability Statement

We explained as follows:

The data that support the findings of this study are available from the corresponding author upon reasonable request through the Japanese Association for Acute Medicine for researchers who meet the criteria for access to confidential data. Data are available from the Japanese Association for Acute Medicine Ethics Committee with the following contact information: E-mail jaam-6@bz04.plala.or.jp.

b) List of the names of all of the ethics committees that reviewed and approved this study

We provided the information as a Supporting Information file.

---

## [Decision Letter · Decision Letter 1]

23 Dec 2019

PONE-D-19-25111R1

Impact of Blood Glucose Abnormalities on Outcomes and Disease Severity in Patients with Severe Sepsis: An Analysis from a Multicenter, Prospective Survey of Severe Sepsis

PLOS ONE

Dear Prof. Kushimoto,

Thank you for submitting your manuscript to PLOS ONE. After careful consideration, we feel that it has merit but does not fully meet PLOS ONE’s publication criteria as it currently stands. Therefore, we invite you to submit a revised version of the manuscript that addresses the points raised during the review process.

Thank you for this interesting paper on an important and incompletely understood topic. Some concerns were raised by the reviewers in the methods and results sections that will require clarification prior to acceptance for publication. Changes in these sections should also be reflected in discussion, where appropriate.

We would appreciate receiving your revised manuscript by Feb 06 2020 11:59PM. To enhance the reproducibility of your results, we recommend that if applicable you deposit your laboratory protocols in protocols.io, where a protocol can be assigned its own identifier (DOI) such that it can be cited independently in the future. For instructions see: http://journals.plos.org/plosone/s/submission-guidelines#loc-laboratory-protocols

We look forward to receiving your revised manuscript.

Kind regards,

Robert Ehrman, MD, MS

Academic Editor

PLOS ONE

Reviewers' comments:

Reviewer's Responses to Questions

**Comments to the Author**

1. If the authors have adequately addressed your comments raised in a previous round of review and you feel that this manuscript is now acceptable for publication, you may indicate that here to bypass the “Comments to the Author” section, enter your conflict of interest statement in the “Confidential to Editor” section, and submit your "Accept" recommendation.

Reviewer #1: (No Response)

Reviewer #2: (No Response)

2. Is the manuscript technically sound, and do the data support the conclusions?

Reviewer #1: Partly

Reviewer #2: Yes

3. Has the statistical analysis been performed appropriately and rigorously? 

Reviewer #1: No

Reviewer #2: Yes

4. Have the authors made all data underlying the findings in their manuscript fully available?

Reviewer #1: Yes

Reviewer #2: Yes

5. Is the manuscript presented in an intelligible fashion and written in standard English?

Reviewer #1: Yes

Reviewer #2: Yes

6. Review Comments to the Author

Reviewer #1: Summary

Thank you for the opportunity to review this paper. This is a very interesting study that aims to elucidate the predictive role of hypoglycemia in septic patients. The authors utilize a large prospective cohort study to answer the question of whether initial hypoglycemia is predictive of outcome. The results are interesting, but I have issues with how the data was analyzed and presented. Hepatic failure needs to be accounted for specifically in the analyses. As it is currently presented, I think this study is at risk of making the assumption that hypoglycemia is predictive of death, when it is actually just associated with severe disease.

Major points

The study objective to elucidate dysglycemia in a sepsis population is clinically relevant

The authors state “initial blood glucose” level but do not provide timing of this initial glucose or how this was obtained. Was this from finger stick measurements, blood chemistries, etc? Also what was the mean time to measurement of this value from enrollment into the study? It is important to know where in the course of their disease the hypoglycemia occurred.

Even though the parent study (FORECAST) was designed prior to Sepsis-3, they should still report the proportion of patients meeting Sepsis-3 definitions. This should not be difficult given that the authors have SOFA score data available and will make the study more directly comparable to more recent studies.

The authors provide data to support for their choice of blood glucose groups. It would be important to know what they learned from their own data, rather than to utilize prior categorizations of blood glucose from other work. Separating the blood glucose measurements of their patients by quartiles (or similar) and looking at outcomes based on this would be informative.

How did the authors handle censored data for SOFA score in patients who died early?

A larger proportion of patients in the hypoglycemia group had glucose mod to severe liver disease. It is not surprising that those patients are more prone to hypoglycemia. Also, the SOFA scores of the hypoglycemia patients are significantly higher. It is important to know what the Hepatic SOFA scores are for the patients with hypoglycemia. The predictive ability of hypoglycemia could be confounded by liver disease or acute liver failure.

Given the potential interaction between liver disease/failure above it is important to add lactate levels (initial and repeat) to Table 1 and compare across groups.

Corticosteroid requirement is much higher in hypoglycemia group. Is there any data on whether the glucose levels were obtained prior to administration of corticosteroids or if these are post-steroid measures?

The presentation of the data is confusing. The authors mention Kaplan-Meier curves for survival, but then use a Cox regression. But they also report unadjusted odds ratios. If the outcome of interest is the association with mortality, I don’t think a Cox regression is needed as the time to death is not really that relevant. I also do not think unadjusted odds ratios should be presented, and should only be added as supplementary material. The Cox regression shows a higher hazard ratio for glucose < 70, which was robust to sensitivity analysis I think components of SOFA, specifically liver failure should be evaluated in the model as well as Charlson score for mod to severe liver disease. I think we are at risk of finding that hypoglycemia is associated with severely ill patients at risk of mortality, rather than predictive of itself.

Why are unadjusted odds ratios displayed in Table 3 and 4, rather than adjusted?

Minor

Would prefer to see the tables separately and not in the body of the text, which makes it harder to read

Also the authors should indicate units appropriately in tables (N, %) in the first column

Odds ratios should only be reported to 1 or 2 places after the decimal

Reviewer #2: An interesting subgroup analysis highlighting the association between hypoglycemia on admission and mortality in severe sepsis patients from a previously performed multicenter prospective cohort. Highlighting an association between hypoglycemia in non-diabetic sepsis patients and increased mortality is a relevant finding.

Appropriate statistics used for the most part and findings correctly described as an association only in need of further study to determine causation.

The fact that the cohort is a mixed ICU population including medical and trauma patients may confound some of the data a bit. Would be interesting to see if the association still stands in both the trauma and medical ICU subgroups or if it would change.

Abstract still has a few errors and needs some rewording to aid in understanding for readers.

7. PLOS authors have the option to publish the peer review history of their article (what does this mean?). If published, this will include your full peer review and any attached files.

Reviewer #1: No

Reviewer #2: No

---

## [Author Response · Author response to Decision Letter 1]

17 Jan 2020

Reviewer #1:

Summary

Thank you for the opportunity to review this paper. This is a very interesting study that aims to elucidate the predictive role of hypoglycemia in septic patients. The authors utilize a large prospective cohort study to answer the question of whether initial hypoglycemia is predictive of outcome. The results are interesting, but I have issues with how the data was analyzed and presented. Hepatic failure needs to be accounted for specifically in the analyses. As it is currently presented, I think this study is at risk of making the assumption that hypoglycemia is predictive of death, when it is actually just associated with severe disease. 

(Response)

We thank the Reviewer for the favorable comments and acknowledgment of our work. Accordingly, we have revised our manuscript. To facilitate the review process, the revised sentences are shown in red in the revised manuscript.

Major points

The study objective to elucidate dysglycemia in a sepsis population is clinically relevant

The authors state “initial blood glucose” level but do not provide timing of this initial glucose or how this was obtained. Was this from finger stick measurements, blood chemistries, etc? Also what was the mean time to measurement of this value from enrollment into the study? It is important to know where in the course of their disease the hypoglycemia occurred. 

(Response)

Thank you for your comments. Accordingly, we have added the following information in the methods and limitations section:

Methods: Blood glucose levels were measured using a blood gas analyzer (and not a glucometer) as recommend by Surviving Sepsis Campaign: International Guidelines for Management of Severe Sepsis and Septic Shock: 2012 (14). All glucose levels were measured prior to the administration of corticosteroids (if required).

Limitations: Third, although we used the blood glucose levels measured during initial evaluation and management, the time between measurement and the diagnosis of severe sepsis or septic shock was not defined in this registry. These measurement related issues may have influenced the results.

Even though the parent study (FORECAST) was designed prior to Sepsis-3, they should still report the proportion of patients meeting Sepsis-3 definitions. This should not be difficult given that the authors have SOFA score data available and will make the study more directly comparable to more recent studies. 

(Response)

Thank you for your comments. We have added the outcome data for patients diagnosed with sepsis by the Sepsis-3 definition in Supplementary Table 2. For defining an acute increase in SOFA scores of ≥2, a baseline SOFA score of 0 was assumed in patients without a diagnosis of any chronic disease, as defined per APACHE II scores. If a chronic disease was present, a baseline SOFA score of 2 was assigned, similar to methods used in a previous study (JAMA. 2016;317:290–300). 

The authors provide data to support for their choice of blood glucose groups. It would be important to know what they learned from their own data, rather than to utilize prior categorizations of blood glucose from other work. Separating the blood glucose measurements of their patients by quartiles (or similar) and looking at outcomes based on this would be informative. 

(Response)

Thank you for your comments. Although we performed our analysis based on pre-defined categorization of blood glucose levels to recognize the importance of hypoglycemia (70 mg/dL or less), it is important to know the findings obtained from our own data. As such, we have presented all outcomes according to quartiles of patient glucose levels in Supplementary Table 3.

How did the authors handle censored data for SOFA score in patients who died early? 

(Response)

Thank you for your comments. The SOFA score was calculated using physiological and laboratory values determined during the initial evaluation. Therefore, we did not handle censored data for SOFA scores.

We have added the following information in the methods section:

The Sequential Organ Failure Assessment (SOFA) score was calculated using physiological and laboratory values during the initial evaluation.

A larger proportion of patients in the hypoglycemia group had glucose mod to severe liver disease. It is not surprising that those patients are more prone to hypoglycemia. Also, the SOFA scores of the hypoglycemia patients are significantly higher. It is important to know what the Hepatic SOFA scores are for the patients with hypoglycemia. The predictive ability of hypoglycemia could be confounded by liver disease or acute liver failure. 

Given the potential interaction between liver disease/failure above it is important to add lactate levels (initial and repeat) to Table 1 and compare across groups. 

(Response)

Thank you for your comments. We have added lactate levels in Table 1 and the hepatic SOFA scores in the main text according to the reviewer’s comments. However, we have no repeated values of lactate levels in this dataset.

Added data and information:

Lactate levels in Table 1: 3.00 (1.80-5.30), 5.50 (3.05- 9.75), 2.70 (1.60- 5.10), 2.80 (1.78- 4.93), 3.25 (2.18-5.60), respectively.

Hepatic SOFA scores in the main text (Results section): The hepatic SOFA scores in the categories with glucose levels of <70, 70-139, 140-179, and >180 mg/dL were 1 (0-1), 0 (0-1), 0 (0-1), and 0 (0-1), respectively (p<0.001). 

We have also added the following comments:

This score was significantly higher in the group with patients with blood glucose levels <70 mg/dL than in the other groups.

Corticosteroid requirement is much higher in hypoglycemia group. Is there any data on whether the glucose levels were obtained prior to administration of corticosteroids or if these are post-steroid measures? 

(Response)

Thank you for your comments. We have added the following information in the methods section: 

All glucose levels were measured prior to the administration of corticosteroids (if required).

The authors mention Kaplan-Meier curves for survival, but then use a Cox regression. But they also report unadjusted odds ratios. If the outcome of interest is the association with mortality, I don’t think a Cox regression is needed as the time to death is not really that relevant. I also do not think unadjusted odds ratios should be presented, and should only be added as supplementary material. The Cox regression shows a higher hazard ratio for glucose < 70, which was robust to sensitivity analysis I think components of SOFA, specifically liver failure should be evaluated in the model as well as Charlson score for mod to severe liver disease. I think we are at risk of finding that hypoglycemia is associated with severely ill patients at risk of mortality, rather than predictive of itself. 

Why are unadjusted odds ratios displayed in Table 3 and 4, rather than adjusted? These should not be displayed though they can be presented in the body of the manuscript, briefly. 

(Response)

Thank you for your comments. The Kaplan-Meier curves for survival were deleted from the main body of the manuscript and presented as supplementary content. Tables 3 and 4 are presented without the unadjusted odds ratios in the main body of the manuscript, and odds ratios are presented in Supplementary Tables 4 and 5.

The reviewer has highlighted the significance of the coexistence of moderate-to-severe liver disease. Although these comments were reasonable, coexisting moderate-to-severe liver disease was only present in 26 patients showing significantly higher mortality and prevalence of hypoglycemia (Mortality: 255/1132, 22.5% vs. 11/26, 42.3%, p=0.011; Hypoglycemia: 61/1132, 5.4% vs 8/26, 30.8%, p<0.001). Based on these interactions, we did not evaluate the coexistence of moderate-to-severe liver disease as an explanatory variable in this study.

In addition, the outcome of interest was not only mortality but also the duration of survival, as demonstrated in the Kaplan-Meier survival curves. Therefore, we evaluated the significance of hypoglycemia using Cox regression analysis. 

Minor 

Would prefer to see the tables separately and not in the body of the text, which makes it harder to read

(Response)

We have revised the manuscript as per your suggestion.

Also the authors should indicate units appropriately in tables (N, %) in the first column

(Response)

We add the (N, %) in the tables as per your suggestion.

Odds ratios should only be reported to 1 or 2 places after the decimal

(Response)

We have revised the description of odds ratios and hazard ratios as per your suggestion.

Reviewer #2: An interesting subgroup analysis highlighting the association between hypoglycemia on admission and mortality in severe sepsis patients from a previously performed multicenter prospective cohort. Highlighting an association between hypoglycemia in non-diabetic sepsis patients and increased mortality is a relevant finding.

Appropriate statistics used for the most part and findings correctly described as an association only in need of further study to determine causation.

(Response)

We thank the Reviewer for the favorable comments and acknowledgment of our work. Accordingly, we have revised our manuscript. To facilitate the review process, the revised sentences are shown in red in the revised manuscript.

The fact that the cohort is a mixed ICU population including medical and trauma patients may confound some of the data a bit. Would be interesting to see if the association still stands in both the trauma and medical ICU subgroups or if it would change.

(Response)

Thank you for your comments. We agree with the Reviewer’s comments. However, the dataset of the FORECAST Sepsis cohort has no information on trauma and medical ICU subgroups.

Abstract still has a few errors and needs some rewording to aid in understanding for readers.

(Response)

Thank you for your comments. We revised the abstract as you kindly suggested. In addition, we revised some sentences according to the changes made in the main manuscript.

---

## [Decision Letter · Decision Letter 2]

4 Feb 2020

PONE-D-19-25111R2

Impact of Blood Glucose Abnormalities on Outcomes and Disease Severity in Patients with Severe Sepsis: An Analysis from a Multicenter, Prospective Survey of Severe Sepsis

PLOS ONE

Dear Prof. Kushimoto,

Thank you for submitting your manuscript to PLOS ONE. After careful consideration, we feel that it has merit but does not fully meet PLOS ONE’s publication criteria as it currently stands. Therefore, we invite you to submit a revised version of the manuscript that addresses the points raised during the review process.

Thank you for your time and effort revising the manuscript, which has been substantially improved. I am in agreement with the reviewer about the potential confounding effect of liver disease--are the poor outcomes related to hypoglycemia itself, or is this simply a manifestation of liver disease? With only 26 patients in this category, I think it would be difficult to make any firm conclusions about which factor drove the outcome (ie, an analysis comparing liver disease with vs without hypoglycemia would be under-powered).

However, I do think it is worth addressing, most substantively in the discussion, as well mentioning this in the abstract. Can you also include in the methods what criteria were used to classify patients as "moderate to severe liver disease"? I think this would also be informative to readers.

Also, I think the first line of the abstract should read "Dysglycemia **is**..." not "Dysglycemia **are**..."

We would appreciate receiving your revised manuscript by Mar 20 2020 11:59PM. To enhance the reproducibility of your results, we recommend that if applicable you deposit your laboratory protocols in protocols.io, where a protocol can be assigned its own identifier (DOI) such that it can be cited independently in the future. For instructions see: http://journals.plos.org/plosone/s/submission-guidelines#loc-laboratory-protocols

We look forward to receiving your revised manuscript.

Kind regards,

Robert Ehrman, MD, MS

Academic Editor

PLOS ONE

Reviewers' comments:

Reviewer's Responses to Questions

**Comments to the Author**

1. If the authors have adequately addressed your comments raised in a previous round of review and you feel that this manuscript is now acceptable for publication, you may indicate that here to bypass the “Comments to the Author” section, enter your conflict of interest statement in the “Confidential to Editor” section, and submit your "Accept" recommendation.

Reviewer #1: All comments have been addressed

2. Is the manuscript technically sound, and do the data support the conclusions?

Reviewer #1: Yes

3. Has the statistical analysis been performed appropriately and rigorously? 

Reviewer #1: I Don't Know

4. Have the authors made all data underlying the findings in their manuscript fully available?

Reviewer #1: Yes

5. Is the manuscript presented in an intelligible fashion and written in standard English?

Reviewer #1: Yes

6. Review Comments to the Author

Reviewer #1: Overall the authors have addressed most of my concerns. However, now that they have identified the significant influence of hepatic disease (elevated hepatic SOFA) in the subgroup with glucose < 70, the reader needs to be sure that hepatic failure isn't driving most of the difference seen in this group. I am not sure based on reviewing their regression models that liver failure was fully accounted for here. It is not explicitly stated nor made clear in the methods. This point may require a statistical review. Moreover, I believe the finding of significantly elevated hepatic SOFA should be mentioned in the abstract, discussion and conclusion.

7. PLOS authors have the option to publish the peer review history of their article (what does this mean?). If published, this will include your full peer review and any attached files.

Reviewer #1: No

---

## [Author Response · Author response to Decision Letter 2]

14 Feb 2020

Editor’s comments

Thank you for your time and effort revising the manuscript, which has been substantially improved. I am in agreement with the reviewer about the potential confounding effect of liver disease--are the poor outcomes related to hypoglycemia itself, or is this simply a manifestation of liver disease? With only 26 patients in this category, I think it would be difficult to make any firm conclusions about which factor drove the outcome (ie, an analysis comparing liver disease with vs without hypoglycemia would be under-powered). However, I do think it is worth addressing, most substantively in the discussion, as well mentioning this in the abstract. 

(Response)

We thank the Editor for the favorable comments and acknowledgment of our work. Accordingly, we have revised our manuscript. To facilitate the review process, the revised sentences are shown in red in the revised manuscript. According to the Editor’s comments, we revised the manuscript as follows.

We have added the following comments to the discussion section:

The liver plays an important role in glucose metabolism, and hypoglycemia may occur due to impaired ability of the liver to increase plasma glucose through gluconeogenesis (31). Hypoglycemia has also been reported as a common manifestation of sepsis patients in cirrhosis (32). Although the hepatic SOFA score in hypoglycemic patients was significantly higher than those in other groups, the presence of "moderate to severe liver disease" as a comorbidity was not different between hypoglycemia and non-hypoglycemia patients in this study. In addition, hepatic SOFA score on admission in survivors was not higher than that in non-survivors, and the odds ratios of hepatic SOFA scores of ≥1 for in-hospital mortality and 28-day mortality were 1.18 (0.89–1.57) and 1.22 (0.90–1.66), respectively. Hypoglycemia may be related to hepatic impairment accompanied by corticosteroid insufficiency, and further study including a larger number of patients is required to elucidate the impact of the relationship between hypoglycemia and hepatic impairment on the outcome of patients with sepsis.

We have also added the following comments in results section of the abstract: 

Of 1158 patients, 69, 543, 233, and 313 patients were categorized as glucose levels <70 (hypoglycemia), 70-139, 140-179, >180 mg/dL, respectively. Both the Acute Physiological and Chronic Health Evaluation II and Sequential Organ Failure Assessment (SOFA) scores on the day of enrollment were higher in the hypoglycemia patients than non-hypoglycemic patients than in those with 70-179 mg/dL. The hepatic SOFA scores were also higher in hypoglycemia patients.

Can you also include in the methods what criteria were used to classify patients as "moderate to severe liver disease"? I think this would also be informative to readers.

(Response)

We have provided the following definitions in the methods section: Severe, moderate, and mild liver disease as comorbidities were defined as follows: severe; cirrhosis and portal hypertension with variceal bleeding history; moderate, cirrhosis and portal hypertension but no variceal bleeding history; and mild, chronic hepatitis (or cirrhosis without portal hypertension) (23).

Also, I think the first line of the abstract should read "Dysglycemia is..." not "Dysglycemia are..."

➠ We have revised as per your suggestion. 

Reviewer #1: 

Overall the authors have addressed most of my concerns. 

However, now that they have identified the significant influence of hepatic disease (elevated hepatic SOFA) in the subgroup with glucose < 70, the reader needs to be sure that hepatic failure isn't driving most of the difference seen in this group. I am not sure based on reviewing their regression models that liver failure was fully accounted for here. It is not explicitly stated nor made clear in the methods. This point may require a statistical review. 

(Response)

We thank the Reviewer for the favorable comments and acknowledgment of our work. Accordingly, we have revised our manuscript. To facilitate the review process, the revised sentences are shown in red in the revised manuscript.

We have added the following comments in the methods and results sections:

Methods:

Hepatic disease/dysfunction, glycemic abnormalities, Sepsis-3, and outcomes

To evaluate the relationship between hepatic impairment and glycemic abnormalities, the presence of moderate to severe hepatic disease as a comorbidity and hepatic SOFA score were compared among groups. In addition, to assess the influence of hepatic impairment on the outcome of hypoglycemia, the presence of moderate to severe hepatic disease and hepatic SOFA score >0 were compared between hypoglycemia and non-hypoglycemia patients. Severe, moderate, and mild liver disease as comorbidities were defined as follows: severe; cirrhosis and portal hypertension with variceal bleeding history; moderate, cirrhosis and portal hypertension but no variceal bleeding history; and mild, chronic hepatitis (or cirrhosis without portal hypertension).

Results:

Influence of hepatic disease/dysfunction on mortality

The hepatic SOFA scores in the patients who were categorized according to their glucose levels of <70, 70-139, 140-179, and >180 mg/dL were 1 (0-1), 0 (0-1), 0 (0-1), and 0 (0-1), respectively (p<0.001). The score was significantly higher in the blood glucose level <70 mg/dL group than in the other groups (Table 1).

In contrast, the presence of "moderate to severe liver disease" as a comorbidity was not different between hypoglycemic and non-hypoglycemic patients (8/69, 151/1089, p =0.719). Also, hepatic SOFA on admission in survivors was not higher than that of non-survivors (p=0.232). Odds ratios of hepatic SOFA score of 1 or more for in-hospital mortality and 28-day mortality were 1.18 (0.89–1.57) and 1.22 (0.90–1.66), respectively.

Moreover, I believe the finding of significantly elevated hepatic SOFA should be mentioned in the abstract, discussion and conclusion.

(Response) 

We have added the following comments in the abstract, discussion, and conclusion sections of the revised manuscript:

Results section of the abstract: 

The hepatic SOFA scores were also higher in hypoglycemia patients.

Discussion:

The liver plays an important role in glucose metabolism, and hypoglycemia may occur due to impaired ability of the liver to increase plasma glucose through gluconeogenesis (31). Hypoglycemia has also been reported as a common manifestation of sepsis patients in cirrhosis (32). Although the hepatic SOFA score in hypoglycemic patients was significantly higher than those in other groups, the presence of "moderate to severe liver disease" as a comorbidity was not different between hypoglycemia and non-hypoglycemia patients in this study. In addition, hepatic SOFA score on admission in survivors was not higher than that in non-survivors, and the odds ratios of hepatic SOFA scores of ≥1 for in-hospital mortality and 28-day mortality were 1.18 (0.89–1.57) and 1.22 (0.90–1.66), respectively. Hypoglycemia may be related to hepatic impairment accompanied by corticosteroid insufficiency, and further study including a larger number of patients is required to elucidate the impact of the relationship between hypoglycemia and hepatic impairment on the outcome of patients with sepsis.

Conclusions

Hypoglycemia may be related to increased disease severity and a higher mortality rate in patients with sepsis, and moreover, was an independent predictor of poor clinical outcome. These relationships were evident only in patients without diabetes. Patients with both hypoglycemia and septic shock had increase mortality rates. Although hypoglycemia may be related to hepatic impairment, further studies are needed to explicate the relationship with respect to the outcomes of patients with sepsis.

---

## [Editor Report · Decision Letter 3]

19 Feb 2020

Impact of Blood Glucose Abnormalities on Outcomes and Disease Severity in Patients with Severe Sepsis: An Analysis from a Multicenter, Prospective Survey of Severe Sepsis

PONE-D-19-25111R3

Dear Dr. Kushimoto,

We are pleased to inform you that your manuscript has been judged scientifically suitable for publication and will be formally accepted for publication once it complies with all outstanding technical requirements.

With kind regards,

Robert Ehrman, MD, MS

Academic Editor

PLOS ONE
---

## [Editor Report · Acceptance letter]

26 Feb 2020

PONE-D-19-25111R3 

Impact of Blood Glucose Abnormalities on Outcomes and Disease Severity in Patients with Severe Sepsis: An Analysis from a Multicenter, Prospective Survey of Severe Sepsis 

Dear Dr. Kushimoto:

I am pleased to inform you that your manuscript has been deemed suitable for publication in PLOS ONE. Congratulations! Your manuscript is now with our production department. 

With kind regards,

on behalf of

Dr. Robert Ehrman 

Academic Editor

PLOS ONE